# Targeting Cell Adhesion Molecules via Carbonate Apatite-Mediated Delivery of Specific siRNAs to Breast Cancer Cells In Vitro and In Vivo

**DOI:** 10.3390/pharmaceutics11070309

**Published:** 2019-07-02

**Authors:** Maeirah Afzal Ashaie, Rowshan Ara Islam, Nur Izyani Kamaruzman, Nabilah Ibnat, Kyi Kyi Tha, Ezharul Hoque Chowdhury

**Affiliations:** 1Jeffrey Cheah School of Medicine and Health Sciences, Monash University Malaysia, Jalan Lagoon Selatan, Bandar Sunway, Subang Jaya 47500, Malaysia; 2Health & Wellbeing Cluster, Global Asia in the 21st Century (GA21) Platform, Monash University Malaysia, Jalan Lagoon Selatan, Bandar Sunway, Subang Jaya 47500, Malaysia

**Keywords:** breast cancer, siRNA, gene silencing, nanoparticle, carbonate apatite, α-catenin, β-catenin, talin-1, vinculin, paxillin and actinin-1

## Abstract

While several treatment strategies are applied to cure breast cancer, it still remains one of the leading causes of female deaths worldwide. Since chemotherapeutic drugs have severe side effects and are responsible for development of drug resistance in cancer cells, gene therapy is now considered as one of the promising options to address the current treatment limitations. Identification of the over-expressed genes accounting for constitutive activation of certain pathways, and their subsequent knockdown with specific small interfering RNAs (siRNAs), could be a powerful tool in inhibiting proliferation and survival of cancer cells. In this study, we delivered siRNAs against mRNA transcripts of over-regulated cell adhesion molecules such as catenin alpha 1 (CTNNA1), catenin beta 1 (CTNNB1), talin-1 (TLN1), vinculin (VCL), paxillin (PXN), and actinin-1 (ACTN1) in human (MCF-7 and MDA-MB-231) and murine (4T1) cell lines as well as in the murine female Balb/c mice model. In order to overcome the barriers of cell permeability and nuclease-mediated degradation, the pH-sensitive carbonate apatite (CA) nanocarrier was used as a delivery vehicle. While targeting CTNNA1, CTNNB1, TLN1, VCL, PXN, and ACTN1 resulted in a reduction of cell viability in MCF-7 and MDA-MB-231 cells, delivery of all these siRNAs via carbonate apatite (CA) nanoparticles successfully reduced the cell viability in 4T1 cells. In 4T1 cells, delivery of CTNNA1, CTNNB1, TLN1, VCL, PXN, and ACTN1 siRNAs with CA caused significant reduction in phosphorylated and total AKT levels. Furthermore, reduced band intensity was observed for phosphorylated and total MAPK upon transfection of 4T1 cells with CTNNA1, CTNNB1, and VCL siRNAs. Intravenous delivery of CTNNA1 siRNA with CA nanoparticles significantly reduced tumor volume in the initial phase of the study, while siRNAs targeting CTNNB1, TLN1, VCL, PXN, and ACTN1 genes significantly decreased the tumor burden at all time points. The tumor weights at the end of the treatments were also notably smaller compared to CA. This successfully demonstrates that targeting these dysregulated genes via RNAi and by using a suitable delivery vehicle such as CA could serve as a promising therapeutic treatment modality for breast cancers.

## 1. Introduction

Breast cancer, which is a group of heterogeneous diseases, is divided into various categories based on characteristics such as histological grades, gene expression, and clinical levels [1]. While research still does not provide the exact mechanism involved in the invasion and metastasis of various types of breast cancers, studies indicate aberrated cell adhesion molecules play a critical role in triggering a cascade of pathways, which are involved with initiation, progression, invasion, and metastasis of tumor cells to other organs [2,3,4,5].

The structural and functional integration of cells with their surrounding cells and matrix takes place via various cell adhesion proteins apart from integrins and cadherins. These proteins can be divided into several functional groups such as structural proteins, adaptor proteins, focal adhesion proteins, cytoskeletal actin-binding proteins, and signaling proteins. Some of the molecules that fall within these groups include catenins, actin, actinin, talin, vinculin, and paxillin. These protein molecules localize to integrin mediated cell-extracellular matrix adhesion contacts as well as cadherin mediated cell-cell contacts, at cytoplasmic domain, bridging the adhesion molecules with cytoskeleton. Apart from playing a role in cellular processes and signal transduction pathways, they also contribute toward various diseases such as cancer [6,7,8,9,10,11,12,13,14,15]. For instance, paxillin, which is a multi-domain adaptor protein, is involved in essential processes such as embryonic development, wound repair, and tumor metastasis by binding to various signaling proteins associated with actin cytoskeleton organization [16]. The dysregulated expression of these molecules have shown to impair the functioning of cell adhesion molecules such as that of cadherins and integrins [17,18]. For example, in basal like breast cancers, increased expression of α-actinin-1 was found to be responsible for cell migration of mammary epithelial breast cancer cells as a result of destabilization of E-cadherin adhesion [18]. The expression of these additional cell adhesion proteins is diverse in breast cancer and influences stages of invasion, proliferation, progression, and metastasis [7,19,20]. Co-localization of vinculin with activated Akt was shown not only to promote tumor cell invasion but also contribute to extracellular matrix stiffness, which, in turn, promoted tumor progression in mammary epithelium [21]. Another study done on tumor samples from primary breast carcinoma showed a presence of paxillin, which is associated with upregulation of HER2. Furthermore, depending on the status of HER2, the response of chemotherapy targeting this adaptor protein varies [22]. From these and other studies, it can be inferred that, besides conventional treatment regimens such as surgery, radiations, and chemotherapy, which are less effective especially in metastatic stages of breast cancer, targeting these additional protein cell adhesion molecules may serve an essential gateway for more effective treatment strategies. Furthermore, the targeting can also enhance the chemosensitivity of the drugs to tumor cells, which, otherwise, show multiple drug resistance. For instance, reduced expression of talin-1 enhanced chemosensitivity of breast cancer cells to docetaxel drug, which proposes talin-1 as a potential therapeutic target [20]. Similarly, various molecules have been identified, which can disrupt dysregulated WNT signaling and catenin dependent transcription. This inhibits the growth of carcinoma cells [23]. In another study, knockdown of β-catenin in triple negative breast cancer (TNBC) cells enhanced the tumor cell sensitivity to doxorubicin and cisplatin drugs, which indicates the essential role of transcriptional activity of this catenin sub-unit for TNBC cell chemosensitivity. Therefore, this makes it a promising therapeutic target for this type of breast cancer [23]. 

Thus, appropriate delivery of antagonistic molecules to regulate expression of these cell adhesion proteins could enhance the efficacy of treatment modalities and control tumor progression. While various molecular-targeted therapies are currently under research, one such approach, which is gaining momentum, is RNA interference (RNAi). Delivery of short double stranded RNA sequences known as small interfering RNAs (siRNAs), which target overregulated genes by inhibiting the gene expression via selective cleavage of an mRNA sequence in the cell cytoplasm, is being explored as a therapeutic option. Figure 1 highlights some of the advantages of siRNA treatment. However, due to limitations of naked siRNA in penetrating plasma membrane and its susceptibility to nuclease mediated degradation [24,25], use of a delivery vehicle to carry and release the siRNA at the target site is required.

While there are various delivery vehicles currently being used, nanoparticles serve as one of the promising vehicles for improved target-specific delivery. In order to design various nanocarriers properties such as size, surface morphology, loading efficiency, and release kinetics, hetero-dynamic properties are taken into consideration [26]. Various biodegradable nanoparticles such as liposomes, polymeric nanoparticles, and inorganic nanoparticles, which have been used as carriers for drug and molecular-based active therapeutic agents, have shown a promising effect by minimizing the side effects as well as increasing the release of the agents to the specific site of action [27]. MM-302, which is a liposome encapsulating doxorubicin, has characteristics for selective uptake of drugs to tumor cells only and is being evaluated in a clinical trial for treatment of advanced metastatic breast cancer, which is positive to HER2 receptors [28]. Similarly, Genexol-PM^®^, a polymeric micelle loaded with paclitaxel drugs, is being used for treatment of breast cancer [29]. In another study, the Au-Fe_3_O_4_ nanoparticle scaffold loaded with cisplatin anti-cancer drugs and the herceptin antibody has been used as targeted delivery to breast cancer cells [30]. Likewise, delivery of microRNA 34a (miR-34a) encapsulated with hyaluronic acid/protamine sulfate nanocapsule has shown apoptosis and cell death in triple negative breast cancer cells/tissues and reduced the proliferation of cancer cells [31].

Recently, pH sensitive inorganic carbonate apatite (CA) is gaining momentum in therapeutic delivery [32,33,34]. Studies show CA is effective in intracellular delivery and release of therapeutic agents such as DNA, siRNAs, and chemotherapeutic drugs from endosomal vesicle to cytosol of the cells, which allows the escape of enzymatic degradation and targeted delivery [35,36,37,38]. Therefore, this serves as a promising therapeutic carrier. Various earlier in vitro studies showed successful reduction of cell viability when CA-siRNA was employed to knockdown a target gene compared to the treatments with free siRNA and CA particles. Furthermore, cellular uptake study showed that the efficiency of siRNA uptake using CA as a vehicle was higher than free siRNAs, which indicates efficient internalization of CA-siRNA complexes. The siRNA-CA complex formation was due to ionic interactions between negative phosphate backbone of siRNA and cation (Ca^2+^)-rich domains of CA nanoparticles. Bio-distribution analysis following 4 hours of intravenous administration revealed high accumulation of siRNA in the tumors when delivered with CA, compared to free siRNA, which virtually showed no accumulation. This indicates the profound role of nanoparticles in delivery of siRNAs to the target site with the consequential effect on the tumor reduction [39,40].

Identifying the essential molecules that play a role in enhancing the tumor proliferation and growth in breast cancer and targeting them by siRNA-based therapy and nanotechnology could serve promising therapeutic avenue against primary and metastatic breast cancers. Therefore, in this work, we have targeted dysregulated additional cell adhesion proteins, namely, catenin alpha 1 (CTNNA1), catenin beta 1 (CTNNB1), talin-1 (TLN1), vinculin (VCL), paxillin (PXN), and actinin-1 (ACTN1) via CA nano-carrier-facilitated delivery of specific siRNAs to investigate their potential therapeutic roles in inhibiting proliferation and survival of breast cancer cells in vitro and in the murine model of breast cancer. By developing these suitable molecular targeted therapies, pharmaceutical advancement in breast cancer treatment could be enhanced.

## 2. Materials and Methods

### 2.1. Materials 

Cell lines: Human breast cancer cell line (MCF-7), TNBC (MDA-MB-231), and murine metastatic breast cancer cell line (4T1). 

Reagents: Dulbecco’s modified eagle medium (DMEM) powder, fetal bovine serum (FBS), penicillin streptomycin, trypLE Express without phenol red and 4-(2-hydroxyethyl)-1-piperazineethanesulfonic acid (HEPES) from Gibco BRL (Carlsbad, CA, USA). DMEM media, calcium chloride dihydrate (CaCl_2_·2H_2_O), sodium bicarbonate (NaHCO_3_), potassium phosphate monobasic (KH_2_PO_4_), sodium phosphate dibasic heptahydrate (H_15_Na_2_O_11_P), bovine serum albumin (BSA), skim milk powder, tris (hydroxmethyl)amino-methane, phosphate inhibitor cocktail 2, EDTA-free protease inhibitor tablets, dimethyl sulphoxide (DMSO), and thiazolyl blue tetrazolium bromide (MTT), methanol, and trypan blue (0.4%) from Sigma-Aldrich (St. Louis, MO, USA). Sodium chloride (NaCl) and potassium chloride (KCl) from Fisher Scientific (Loughborough, Leicestershire, UK). Sodium dodecyl sulfate (SDS) and Restore PLUS Western blot stripping buffer from Thermo Fisher Scientific (Rockford, IL, USA). Triton C-100, glycine, blotting-grade blocker, Clarity Western ECL substrate, tween-20, bromophenol blue, quickstart Bradford 1X dye reagent, quickstart bovine serum albumin (BSA) standard set, dithiothreitol (DTT), and mini-protean TGX gels from Bio Rad (Hercules, CA, USA). Validated siRNAs from Qiagen (Valencia, CA, USA).

### 2.2. Methodology

#### 2.2.1. Cell Culture

MCF-7, MDA-MB-231, and 4T1 cell lines were cultured in 75 cm^2^ tissue flasks (Nunc, Orlando, FL, USA) with DMEM supplemented with 10% FBS, 50 μg/mL penicillin, 50 μg/mL streptomycin, and 1% HEPES, at 37 °C in humidified 5% CO_2_ incubator.

#### 2.2.2. siRNA Design and Sequence

All the siRNAs were validated by Qiagen. Table 1 lists the siRNAs used and their sequences. Additionally, 1 nmol lyophilized power of each of the siRNAs targeting CTNNA1, CTNNB1, TLN1, VCL, PXN, and ACTN1 was reconstituted with RNase-free water to obtain 10-μM stock solution and stored at –20 °C.

#### 2.2.3. Infrared Spectroscopy

Fourier transform-infrared (FT-IR) spectroscopy of generated carbonate apatite particles was performed using Varian 440 FTIR (Santa Clara, CA, USA). CA nanoparticles were prepared as discussed earlier [41]. Additionally, 44mM of sodium bicarbonate and DMEM powder was mixed using mili-Q water with pH adjusted to 7.4. 40 mM of Ca was added and, in order to generate CA nanoparticles, the mixture was incubated at 37 °C for 30 min. After 30 min, the complex was centrifuged at 3700 rpm, 4 °C for 30 min. After 30 min, the supernatant was removed and pellet was washed with mili-Q water and re-centrifuged under the same conditions. This was followed by removal of supernatant and in addition to a sufficient amount of mili-Q water to cover the pellet. The sample was left overnight in 4 °C. After overnight incubation at 4 °C, the sample was freeze dried and the powder proceeded for elemental analysis.

#### 2.2.4. Generation of CA-siRNA Complex for In Vitro Study 

CA nanoparticles were prepared as discussed earlier [41]. Briefly, 44 mM sodium bicarbonate and DMEM powder was mixed using mili-Q water, with pH adjusted to 7.4. To the final 1 mL volume of this media, 1 nM of siRNA against CTNNA1, CTNNB1, TLN1, VCL, PXN, and ACTN1 was added individually, which was followed by the addition of 4 μL of 1M CaCl_2_. The final mixture was incubated at 37 °C for 30 min to generate specific siRNA-CA complexes. After 30 min of incubation, 10% FBA was added to stop the formation of complexes and prevent particle aggregation.

#### 2.2.5. Cell Proliferation Assay by 3-(4,5-Dimethylthiazaol-2-yl)-2,5-diphenyltetrazolium Bromide (MTT)

In order to assess cytotoxicity of siRNA-CA complex on MCF-7, MDA-MB-231, and 4T1 breast cancer cell lines, the MTT assay was performed. Briefly, cells from the exponential phase were seeded in a 24-well plate at a density of 50,000 cells/well in complete DMEM media (DMEM with 10% FBS, 1% PenStrep, and 1% HEPES) and the plate was incubated for 24 h at 37 °C in a humidified environment of an incubator with a 5% CO_2_ supply. After 24 h, the seeded cells were treated with prepared siRNA-CA complexes and incubated again at 37 °C in the incubator for 48 h. After 48 h, 50 μL of MTT solution (5 mg/mL) was added to each well and the plate was incubated for 4 h. Afterward, the media was removed and 300 μL of DMSO was added to each well prior to incubation for 5 min to completely dissolve the purple formazan crystals. Absorbance was taken using a microplate reader (Bio-Rad) at an optical density of 595 nm with a reference wavelength of 630 nm. 

#### 2.2.6. SDS-PAGE and Western Blot 

Prior to proceeding with in vivo tumor regression assay, protein expression of 4T1 cells treated with siRNA-CA complexes was checked using the Western blot. 4T1 cells were seeded and treated as mentioned earlier. After 48 h, lysis was done using a lysis buffer containing 20 mM Tris-HCl (pH 7.5), 150 mM NaCl, 1 mM EDTA, 50 mM beta-glycerolphosphate, phosphate inhibitor cocktail 2 1 mM EGTA, 50 nM NaF, complete EDTA-free protease inhibitor, and 1% Triton X-100. The lysates were collected and centrifuged at 13,000 rpm for 8 min, before collecting the supernatant and discarding the pellet. The Quick Start Bradford Protein Assay kit was used to determine lysate protein concentrations. Additionally, 9 μg proteins were loaded per well in precast gels, and the SDS-PAGE was run, which was followed by turbo transfer of proteins from gel to the nitrocellulose membrane. The nitrocellulose membranes were blocked for 1 h at room temperature with 5% non-fat dry milk in 1X TBST (pH 7.4, containing 0.1% Tween 20) and probed with a primary antibody overnight at 4 °C. After overnight probing, blots were washed with 1X TBST, which was followed by horseradish peroxide-conjugated secondary antibody probing at room temperature for 1 h. Excess of the secondary antibody was removed by washing membranes five times with 1X TBST. The Clarity Western ECL substrate was used to detect the signals. Membranes were visualized by using a Bio-Rad Gel Document System (Bio Rad). Densitometry analysis was done using ImageJ software (ImageJ 1.52a, Wayne Rasband, National Institutes of Health, USA, 2018) and treatment groups were normalized with CA treated controls.

#### 2.2.7. In Vivo Tumor Regression Analysis

In vivo work was done according to the instructions of Monash University Animal Ethics Committee, which approved the protocol for animal experiments (MARP/2016/126, 06/12/2016). Six to eight week-old female Balb/c mice of 15–20 gm body weights (obtained from School of Medicine and Health Science Animal Facility, Monash University, Subang Jaya, Malaysia) were provided with ab libitum and water and maintained in 12:12 light: dark condition throughout the experiment. Approximately 1 × 10^5^ 4T1 cells prepared in 100 µL PBS were subcutaneously injected on the third mammary pad of mice. In order to locate the tumors and measure the size, the area was shaved with hair removal cream. Once tumors reached a palpable stage, 5 mice were randomly arranged in each group. CA nanoparticle complexes were formed with 8 μL of CaCl_2_ in 200 μL of DMEM media supplemented with 44 mM bicarbonate and 3.4 µg/kg of single siRNAs (CTNNA1, CTNNB1, TLN1, VCL, PXN, and ACTN1). The prepared siRNA-CA complexes were incubated at 37 °C for 30 min and then put on ice to slow down the particle aggregation. Additionally, 100 μL of siRNA-CA complexes were injected intravenously at the right or left caudal vein of mice. The treatment was given at the frequency of 4 doses, with two days apart from the previous dose. The vernier caliper was used to measure tumor volume. At the end of the treatment, tumors were excised, weighed, and images were taken.

### 2.3. Data Analysis

Data from the MTT assay was presented as mean ± SD. The cell viability in the treated wells was expressed as a percentage and was calculated using the absorbance values obtained from the MTT assay by the following formula.

% of cell viability = (*OD treated − OD reference*)/(*OD untreated − OD reference*) × 100

The cell death and actual cytotoxicity were calculated by the following formula. 

% cell death = 100% of cell viability 

% Actual cytotoxicity (of siRNA) = (% CA + siRNA cytotoxicity) − (% CA cytotoxicity)

For tumor regression, the analysis tumor volume was calculated using the length and width of the tumor with the following formula. 

Tumor volume (in mm^3^) = *Length* × (*Width*)^2^/2

Statistical analysis for the densitometry was done using a Student *t-*test and in vivo tumor regression analysis was done using SPSS. For an in vivo study, the LSD post-hoc test for one-way ANOVA was used to compare the significant difference between siRNA-CA treated and CA treated groups. Data was considered statistically significant when * represented *p* < 0.05.

## 3. Results 

### 3.1. Elemental Analysis of CA Nanoparticles Using FT-IR Spectroscopy

The formation of CA from the lyophilized sample was confirmed via FT-IR spectroscopy. The IR spectra was collected between 400–3800 cm^−1^ (Figure 2). Three main chemical groups synthesized are hydroxyl (OH^−^), carbonate ion (CO_3_^−^), and phosphate ion (PO_4_^3−^). From the IR spectrum, the OH^−^ stretch can be observed from 3727 to 2946, 1658, and 675 cm^−1^. The peaks that represent CO_3_^−^ can be seen at 1480, 1415, and 866 cm^−1^. The peaks that represent PO_4_^3−^ can be seen at 1008, 585, 567, and 540 cm^−1^ while peaks within 467 cm^−1^ represent weak PO_4_^3−^. Figure 2b shows the magnified image of the essential peaks of CO_3_^−^ and PO_4_^3−^.

### 3.2. Assessment of siRNA Concentration with/without CA-Assisted Delivery in Breast Cancer Cells via the MTT Assay

In order to see the optimum siRNA concentration for cell transfections, the MTT assay was performed where two different cell adhesion siRNAs were used in MCF-7 and 4T1 cells. Three different concentrations of siRNAs were used (10 pM, 100 pM, and 1 nM) with/without CA as a delivery vehicle. From Figure 3a,b, we can see that, compared to free ACTN1 and TLN1 siRNAs, siRNAs bound to CA nanoparticles caused more reduction in cell viability. Furthermore, the reduction in cell viability was higher at a 1-nM concentration of siRNA (~67%).

### 3.3. Role of Additional Cell Adhesion Molecules in Proliferation and Survival of Breast Cancer Cells using the MTT Assay 

Treatment of MCF-7, MDA-MB-231, and 4T1 cells by targeting CTNNA1, CTNNB1, TLN1, VCL, PXN, and ACTN1 genes via siRNA-CA delivery showed varied cell viabilities, based on the MTT assay. Table 2 shows actual cytotoxicity of various treatment groups against three cell lines after 48 h of treatment.

Transfection of CA loaded CTNNA1 and CTNNB1 siRNAs caused slight cell viability reduction in 4T1 cells (Figure 4c). The reduced viability was ~80% for CTNNA1 and ~77% in CTNNB1 siRNA transfection. 

On the other hand, transfection of TLN1 siRNA loaded with CA nanoparticles caused reduced cellular viability by ~67% in MCF-7 cells (Figure 5a) and ~72% in 4T1 cells (Figure 5c). However, in MDA-MB-231 cells, no reduction in cell viability was observed (Figure 5b).

Transfection of VCL siRNA loaded with CA nanoparticles caused no reduction in MCF-7 cell viability compared to a CA nanoparticle treated control (Figure 6a). On the other hand, in 4T1 cells, a mild reduction to ~77% (Figure 6c) could be seen. In MDA-MB-231, ~83% cell viability was observed (Figure 6b). 

Targeting PXN via transfection of CA loaded PXN siRNA caused significant reduction in all three cell lines. The viability greatly reduced in MCF-7 (~57%) and 4T1 (67%) cells (Figure 7a,c, respectively). In MDA-MB-231 cells, viability reduction to ~83% was seen (Figure 7b).

Lastly, transfection of MCF-7 and 4T1 cells with ACTN1 siRNA loaded with CA nanoparticles also caused a decrease in cell viability. The viability after 48 h was ~67% in MCF-7 cells (Figure 8a) and ~72% in 4T1 cells (Figure 8c). However, in MDA-MB-231 cells, the cell viability did not show such a high reduction compared to the CA control (Figure 8b). 

### 3.4. Effects on PI3-Kinase/AKT and MAPK Pathways in 4T1 Cells

The Western blot analysis showed reduced expression of both p-AKT and total AKT proteins, upon 48 h transfection targeting ACTN1, PXN, and CTNNA1 via siRNAs loaded with CA nanoparticles (Figure 9). The reduction was significantly different (*p* < 0.05) compared to CA (Figure 9b,c). However, it was only CTNNA1, CTNNB1, and VCL siRNA transfection that caused low band intensities for p-MAPK and total MAPK (Figure 8a) with *p* < 0.05 (Figure 9d,e).

### 3.5. Delivery of CA Loaded siRNAs against Cell Adhesion Molecules in a Murine Breast Cancer Model

#### 3.5.1. Body Weight 

The average body weights for mice in all the treatment and control groups were monitored at the interval of every two days, and it was observed that the weight in all the groups remained unchanged (approximately 20 g). Data is shown, in this case, for treatments with CA loaded TLN1 (Figure 10a), VCL (Figure 10b), and ACTN1 (Figure 10c) siRNAs.

#### 3.5.2. Tumor Regression of CA Loaded siRNAs in a Mouse Breast Cancer Model

In vivo tumor regression showed reduced tumor volume in the CA-CTNNA1 siRNA complex. The reduced tumor volume was observed after the first dose of treatment (Figure 11a). The highest fold (3.75) reduction was observed on Day 9 (Table 3). * represents a significant reduction with *p* < 0.05, which was observed from Day 9 until Day 17 of treatment (Figure 11a). Post day 17, a significant increase in tumor volume was observed, which, on Day 21, was higher than the CA control. Figure 11b shows an excised tumor of the CA-CTNNA1 siRNA treated group vs. the CA treated group, while Figure 11c shows tumor weight at the end of treatment. The tumor appears to be of the same size and weight compared to CA. 

On the other hand, delivery of the CA-CTNNB1 siRNA complex caused a significant reduction (*p* < 0.05) at all the time points of measurement after injection of the first dose of treatment. Figure 12a shows that the tumor regression curve was almost consistent with no increase until Day 17. The tumor fold reduction in comparison to the CA control (4.80) was highest on Day 10 and Day 21 (Table 4). Figure 12b shows the reduced tumor at the end of the treatment compared to the CA group and decreased tumor weight of ~0.41 g (Figure 12c), which is significantly lesser (*p* < 0.05) when compared to CA.

Treatment of 4T1 induced mice with CA-TLN1, CA-VCL, CA-PXN, and CA-ACTN1 siRNA complex also caused reduced tumor volume compared to the CA treated control. While the trend of tumor volume over period of time kept increasing, the tumor volume was significantly lower (*) than CA control with *p* < 0.05 (Figure 13a, Figure 14a, Figure 15a and Figure 16a, respectively). The highest tumor fold decrease of 2.58 on Day 12 in case of CA-TLN1 was observed (Table 5). Figure 13b shows a reduced tumor of the treatment group at the end of the study with tumor weight being ~0.29g, which is significantly less (*p* < 0.05) than the CA control (Figure 13c). In case of the CA-VCL siRNA treated group, tumor regression volume was significantly lower (*) from Day 12 onwards with *p* < 0.05 (Figure 14a). Until Day 12, the tumor regression trend did not increase, while, from Day 12 onwards, it increased significantly slower than the control. The highest tumor fold reduction was observed on Day 12 with a 3.61-fold decrease (Table 6). Figure 14b shows a reduced tumor for the treatment compared to CA at the end of the study with tumor weight reduced to ~0.24g (Figure 14c), which is significantly less (*p* < 0.05) than CA at ~0.63 g. Compared to the CA control, in PXN, the siRNA-CA treated group tumor regression curve remained constant throughout the experiment. While the tumor burden was ~180 mm^3^ on Day 21 for the control, in the PXN-CA treatment group, the tumor volume did no increase beyond ~30mm^3^. The highest fold decrease was recorded as 6.19 on Day 19 (Table 7). Figure 15b shows a highly reduced tumor at the end of the treatment with average tumor weight being ~0.27 g (Figure 15c), which is significantly less (*p* < 0.05) than CA. Lastly, the ACTN1 siRNA-CA group showed a highly significant reduction in tumor size (*p* < 0.05) on all the days (Figure 16a). Compared to the CA group, the tumor volume curve remained constant up to Day 17, after which it slightly increased with the tumor burden not increasing above ~82 mm^3^. As shown in Table 8, the maximum fold reduction of 4.64 was seen on Day 13 of the tumor regression measurement. Figure 16b shows a reduced tumor volume at the end of the treatment and the average weight of tumors was ~0.20 g (Figure 16c), which is significantly less (*p* < 0.05) than the CA.

## 4. Discussion

Various additional cell proteins play roles in localizing adhesion contacts of cadherins and integrins with actin filaments in the cytoplasm and communicate essential signal transduction. Their aberration plays a critical role in disrupting the signaling pathways and results in various diseases including breast cancer. Some of the essential molecules, which are critically involved in breast cancer, are catenins, actinin, talin, vinculin, and paxillin. Table 9 lists some of the essential pathways these molecules are linked with in various forms of breast cancer. Acting as intermediate molecules, these adhesion proteins are responsible for the cascade of metastatic pathways (Figure 17). Targeting these additional proteins could serve an essential therapeutic approach for breast cancer. However, in order to ensure the enhanced target delivery, the use of a suitable carrier vehicle is one of the key factors. In vitro and in vivo RNAi approaches are being applied to target these endogenous overexpressed genes using CA as a delivery vehicle for a release of siRNA at the target site. Furthermore, to see the effect of knockdown of the essential cell adhesion molecules, we performed a Western blot assay where we targeted AKT and MAPK pathways. Since AKT and MAPK pathways are predominantly responsible for regulating proliferation and survival of cancer cells, downregulation of the pathways also correlate with the induced cytotoxicity of the breast cancer cells.

CA is an inorganic pH sensitive nanocarrier that is formed by a chemical reaction between Ca^2+^, PO_4_^3−^, and HCO_3_^−^, which give rise to an apatite structure with a molecular formula of Ca_10_(PO_4_)_6−*x*_(CO_3_)*_x_*(OH)_2_ [34]. The formation of CA was confirmed by FT-IR, which indicates it as a hydroxyapatite. Via IR spectroscopy on lyophilized CA samples, apart from OH^−^, the crystalline structure is composed of CO_3_^−^ and PO_4_^3−^ (Figure 2a). Three distinct peaks at 1480, 1415, and 866 cm^−1^ can be observed, which represent CO_3_^−^ ion while peaks at 1008, 540–585 cm^−1^ represent PO_4_^3−^ (Figure 2b). Since the CO_3_^−^ ion has four different vibrational modes, the peaks in the range of 1480 to 1415 cm^−1^ are intense peaks due to high energy ν_3_ vibrations. The peak that is formed at 866 cm^−1^ is a weak peak formed due to a low energy ν_2_ vibrational mode. On the other hand, the PO_4_^3−^ ion has an intense ν_3_ band at 1008 cm^−1^ while sharp intense ν_4_ bands are observed in the range of 600 to 540 cm^−1^. Furthermore, weak bands corresponding to ν_2_ vibration are observed in the region of 467 cm^−1^ [42,43,44,45]. Additionally, the position of the CO_3_^−^ peaks at the ν_2_ mode in the hydroxyapatite lattice depends on whether CO_3_^−^ is substituted for the PO_4_^3−^ or OH^−^ ion [46]. As compared to P-O stretching, the O-H stretching due to hydroxyapatite stoichiometry is considered weaker. The adsorbed water band is very wide (3700 to 2800 cm^−1^) with a weaker intensity, which may have resulted from a decrease of OH^−^ ions in the spectrum due to CO_3_^−^ vibrational modes. The peak at 1658 and 675 cm^−1^ is also a characteristic of the OH^−^ bending vibration in water. Several peaks could have been due to removal of water from the hydroxyapatite [39,42,44]. Since the hydroxyapatite is a main mineral component of tissues such as bones, teeth, and cartilage, the generation of this less crystalline CA may not be foreign in the organism’s physical environment, which imposes less of a threat to them [32,47]. 

Prior to cell adhesion siRNA treatments, determination of the suitable siRNA concentration for successful in vitro transfection is required. For this, we treated MCF-7 with a CA-ACTN1 siRNA and 4T1 cells with CA-TLN1 siRNA complexes prepared with 4 mM Ca^2+^ at three different concentrations (10 pM, 100 pM, and 1 nM) of the siRNAs. The two siRNAs were randomly selected to maintain the heterogeneity of the experiments. From Figure 3a,b, two observations could be made. When the siRNAs were delivered without a carrier to the cells, no major reduction in cell viability could be seen, which could be due to two reasons. First, siRNA is anionic in charge with a strong phosphate backbone and is hydrophilic in nature. Hence, it faces repulsion due to anionic cell membranes, which limit its passive delivery across the cell membrane. Second, within physiological conditions, siRNAs are unstable and can be easily degraded by nucleases in plasma and cells [24]. Thus, naked delivery of siRNAs cannot cause sufficient reduction in cellular proliferation. On the contrary, when the same siRNAs were loaded with CA nanoparticles, reduction in the cell viability was observed, which indicates efficient siRNA delivery into the cytoplasm, silencing the overexpressed genes. The effective role of CA as a carrier vehicle of siRNA has been shown in previous in vitro and in vivo studies [39,40]. Significant uptake of siRNA observed in cancer cell lines as well as in tumor tissues compared to free siRNA was correlated to the reduced cell viability and enhanced tumor regression, respectively. The inefficiency of naked siRNA following systemic administration was due to its nuclease-mediated degradation as well as elimination via renal clearance. Furthermore, no changes in cytotoxicity and tumor regression following the scrambled siRNA treatment has been observed in previous studies, which indicates that the enhancement in cytotoxicity and tumor regression is not due to the non-specific interactions of the siRNA with other mRNA transcripts or any other off-target effects of the particular siRNA [39,58]. In addition, CA is highly efficient in the knockdown of target genes even with a picogram amount of the initial siRNA concentration [35]. CA with its cation-rich domains can form a complex with anionic siRNAs and the resultant complex is able to cross the plasma membrane through endocytosis. Furthermore, the complex protects the loaded siRNA from the nuclease attack. Due to being small in size, CA loaded siRNAs can undergo efficient endocytosis, which is followed by low pH-dependent fast dissolution of the particles and consequential endosomal escape of the released siRNAs, which likely occurs by neutralizing the endosomes and disrupting them via osmotic pressure. This prevents the lysosomal degradation [32,34,35,59]. This, in turn, results in the release of the siRNAs in cytosol of the target cell. The second observation from Figure 3a,b was irrespective of the cell lines used. The maximum reduction in cell viability was observed at 1nM siRNA concentration loaded with CA nanoparticles. CA transports the therapeutics via enhanced permeability and a retention effect (EPR effect). As tumor cells consume high oxygen, fast angiogenesis occurs, causing weak vasculature and lymphatic circulation in tumor tissues. Passive targeting transportation of therapeutic materials via CA to tumor cells and interstitum take place where openings of these leaky tumor capillaries release the therapeutics at target tumor sites [60].

Catenins play a prominent role in breast cancer by modulating the tissue integrity by destabilizing the E-cadherin/catenin complex. Studies show varied expression of α-catenin and β-catenin in both primary tumors and metastatic tumors [19,61]. Targeting these molecules in appropriate breast cancer cells may disrupt the E-cadherin/catenin adhesion link and reduce the tumor burden. Based on the MTT assay, we observed that transfection of CA loaded with 1 nM concentration of single CTNNA1 and CTNNB1 siRNAs caused significant reduction in cell viability in 4T1 cells (Figure 4c) with an average actual cytotoxicity of ~13% (CA-CTNNA1 siRNA) and ~16% (CA-CTNNB1 siRNA) (Table 2). An earlier report revealed high expression of α-catenin and β-catenin in metastasized tumors with their high accumulation found in cytoplasm [19,62]. Since the 4T1 model represents the human metastatic breast cancer, upon targeting these genes using the siRNA-CA complex, a significant reduction in cell proliferation could be achieved. Furthermore, when intravenous delivery of CTNNA1 and CTNNB1 siRNAs loaded with CA nanoparticles was done in the 4T1 tumor bearing the Balb/c mice model, the reduced tumor volume compared to the CA nanoparticle treated controls (Figure 11a and Figure 12a) indicated the role of these genes in tumorigenesis. The significant reduction for CA-CTNNA1 siRNA delivery (Figure 11a) was observed with *p* < 0.05 on Day 10, Day 13, and Day 17 of measurements compared to the CA control. Toward the end of the treatment, no significant difference was observed when compared to CA treatment. This may be due to other catenins playing an essential role in cell adhesion [5]. An in vivo tumor regression study showed a significant reduction in the tumor burden on all the days of measurement after the first dose of treatment for CTNNB1-CA delivery (Figure 12a). Furthermore, the tumor volume curve almost remained uniform throughout the experimental procedure, which indicates the essential role of CTNNB1 siRNA knockdown in this cancer model. Catenins are involved in proliferation via various signaling pathways including PI3K/AKT, Ras/MAPK, Wnt, and Hedgehog [62]. Since the pathways such as PI3K (AKT) and MAPK are significantly upregulated in metastatic cancers, we performed the Western blot to see the effect at a protein level after intracellular delivery of CTNNA1 and CTNNB1 siRNAs using CA nanoparticles in 4T1 cells. Our results showed a significantly reduced expression in the levels of phosphorylated and total AKT/MAPK (Figure 9), which indicates effective perturbation of these essential signaling pathways in 4T1 cells via CA loaded siRNAs targeting CTNNA1 and CTNNB1 mRNA transcripts.

Talin (TLN) serves one of the most important focal adhesion proteins, which recruits various adaptor proteins for cell adhesion [63] and its expression has been found to be high in various cancers [63,64]. Targeting this protein might enhance the therapeutic efficiency by downregulating tumor migration, invasion, and metastasis. We found that talin-1 siRNA (TLN1) electrostatistically complexed with CA nanoparticles showed high cytotoxicity in MCF-7 cells as well as 4T1 cells. As the viability was reduced to ~ 67% in MCF-7 cells (Figure 5a) and ~72% in 4T1 cells (Figure 5c), the actual cytotoxicity reached ~30% in MCF-7 cells and ~17% in 4T1-cells (Table 2). These results suggest that silencing TLN1 expression could regulate the cell viability in certain breast cancers. The study showed that the overexpression of TLN1 in cells such as prostate cancer cells could lead to increased signaling of AKT and FAK pathways [13]. We also observed that, in 4T1 cells, the level of expression of both phosphorylated and total AKT was significantly reduced upon siRNA transfection when compared to controls (Figure 9). However, on the other hand, the expression level for phosphorylated and total MAPK was significantly higher, which suggests that other oncogenic pathway(s) or downstream molecules might contribute to the upregulation of this pathway. Our in vivo study showed that delivery of the CA-TLN1 siRNA complex could reduce the tumor burden in the 4T1 induced murine breast cancer model (Figure 13a). The slower tumor growth and lower tumor weight (Figure 13c) in the treatment group correlates with reduced TLN1 expression. We could see that the tumor growth increased very slightly from Day 8 of treatment until Day 17 of treatment, with maximum tumor volume reaching to ~180 mm^3^ on Day 21. By reducing the expression level of TLN1, integrin-talin interaction could be hampered, which disrupts the association of various other signaling proteins and, consequently, reduces AKT/FAK signal transduction. Since this could affect the tumor proliferation rate, we observed only a slight increase in the tumor volume compared to the control. In addition, this might have led to a reduced tumor invasion and metastasis to other sites [13]. In MDA-MB-231 cells, no reduction in viability was observed (Figure 5b). Targeting TLN1 was shown to reduce invadopodium formation, tumor invasion, and metastasis in case of MDA-MB-231 cells [64]. These changes might have taken place in our treatment without any visible effect on cell viability. No change in cell viability could be due to varied conditions of cell growth, low siRNA dosage for this aggressive cell line, or upregulation of another isoform of talin, talin-2 whose role has been found in certain cancers [65].

VCL siRNA-loaded CA caused reduced cell viability in both the aggressive forms of breast cancers (MDA-MB-231 and 4T1). While the cell viability was ~77% in 4T1 cells (Figure 6c), in MDA-MB-231, it was mildly reduced to ~83% (Figure 6b). The actual cytotoxicity was ~9% in MDA-MB-231 cells and ~13% for 4T1 cells (Table 2). A subsequent tumor regression study in the animal model showed significantly lower tumor volume compared to the CA nanoparticle treated group (Figure 14a), with the static tumor growth observed up to Day 12, followed by a slow increase, which reaches a maximum up to 200 mm^3^ on Day 21 in treatment group vs. ~350 mm^3^ in the control. The tumor weight at the end of the treatment was ~0.24 g while, in the CA-treated group, it reached up to ~0.63 g (Figure 14c). VCL is an essential cell adaptor protein that functionally regulates cadherin-based cell-cell adhesion. Upregulated in primary invasive cancers, it mediates force-induced tumor cell invasions by facilitating contractile force in three-dimensional ECM as well as by impeding cell migration in a two-dimensional manner [14,15,53,66]. It also regulates survival and motility via the ERK/MAPK pathway [54]. Our Western blot analysis in 4T1 cells demonstrated that intracellular delivery of CA nanoparticles loaded VCL siRNA caused a significant reduction in both total and phosphorylated AKT and MAPK levels compared to controls (Figure 9), which indicates the successful aberration of oncogenic pathways via knockdown of VCL adhesion molecule. Co-localizing at the invasive borders in human breast tumors and providing ECM stiffness, it associates with growth factor molecules and active AKT, and further facilitates PI3K and growth factor signaling [21].

Our results showed the PXN-CA complex caused high reduction in cell viability in MCF-7 and 4T1 cell lines, with a slight decrease in MDA-MB-231. The viability was ~57% in MCF-7 (Figure 7a), ~83% in MDA-MB-231 cells (Figure 7b), and ~67% in 4T1 cells (Figure 7c). Subsequently, the cytotoxicity of ~30% (MCF-7), ~13% (MDA-MB-231), and ~21% (4T1) was observed (Table 2). Western blot showed significant reduced expression of phosphorylated and total AKT upon delivery of the PXN-CA complex (Figure 9), which indicates the direct impact of the knockdown of the PXN protein on the AKT pathway. However, the expression level of p-MAPK and MAPK was increased. TGF-β could cause activation of the MAPK signaling cascade. Furthermore, since the PXN family consists of two other proteins, which are closely related, they could have also enhanced the expression of MAPK [16,56]. In the tumor regression study, the tumor burden was reduced to a greater extent in the PXN-CA treated group (Figure 5a) compared to the control with an average tumor weight reducing to ~0.24g (Figure 5c), which is around 2.6 folds lesser than the average weight of CA tumors. Throughout the experimental procedure, the tumor growth curve was almost a straight line with the highest tumor volume less than 50 mm^3^ at the end of 21 days. This indicates the importance of targeted PXN knockdown in 4T1 induced breast cancers. As shown in earlier studies, in breast tumors, PXN is overexpressed and regulates breast cancer cell metastasis. With PXN being a downstream target of several activated hormones and growth factors, it associates with aggressive cancers and reduces the survival rate of patients. Therefore, by regulating its expression, it could serve as an essential therapeutic molecule to inhibit breast cancer cell invasion, migration, and metastasis as well as plasticity control [7,16,56]. 

Intracellular delivery of the actinin-1 (ACTN1) complexed with CA in breast cancer cells showed a significant reduction of viability in MCF-7 and 4T1 cell lines, while, in MDA-MB-231, no such effect was observed. The cell viability was reduced to ~66% in MCF-7 (Figure 8a), ~87% in MDA-MB-231 (Figure 8b), and ~ 71% in 4T1 cells (Figure 8c). The cytotoxicity was ~26% in MCF-7 cells, ~6% in MDA-MB-231 cells, and ~18% in 4T1 cells (Table 2). This indicates the important role of this gene in proliferation of breast cancer cells. The slight decrease in viability of MDA-MB-231 could be due to higher invasive properties of these cells. Furthermore, upregulated α-Actinin-4 might also enhance the cell viability, as this form of actinin was shown to enhance cell proliferation in various cancers [57]. The Western blot showed significantly reduced expression in phosphorylated and total AKT levels upon transfection of 4T1 cells (Figure 9), which are upregulated in overexpressed ACTN1 cells. No reduction in p-MAPK and MAPK could either be due to the presence of α-Actinin-4, which could activate the ERK/MAPK pathway, or due to cross talks of this pathway with other biological cell molecules, cell receptors, and transcription factors. In addition, both actinin forms have been found to be linked with EGFR, where EGFR causes stimulation of actinin phosphorylation (actinin-4 more than actinin-1), which activates the cascade of signaling pathways [67]. An in vivo tumor regression result was correlated to the cell viability data obtained in 4T1 cells and showed a highly significant reduction in tumor volume upon delivery of ACTN1-CA complex (Figure 16a) compared to the CA control. The tumor growth curve for the treatment group was almost constant until Day 17 of treatment, which indicates controlled tumor growth, and, after Day 17, a slight increase was observed. The tumor volume did not increase more than ~80 mm^3^, with tumor weight not exceeding 0.24 g (Figure 16c). In tumors, α-Actinin-1 serves as an adapter protein, which, under pressure, redistributes to membranes where it facilitates recruitment of the Src protein to β1-integrin-associated focal adhesion, while the other unit, α-Actinin-4, has been focused upon for its tumorigenesis role in breast cancer. ACTN1 still remains a potential molecule to be explored for its contribution toward the tumor burden. Therefore, by targeting this protein, it could provide a therapeutic model for inhibition of pressure stimulated tumor cell adhesions and control of the tumor growth and metastasis [11,68,69,70,71]. 

In the treatment groups, the high reduction in tumor volume might also have been due to frequent 4 doses of administration, which allows the siRNA-CA complex to last for a longer duration in the system, before being degraded and eliminated from the system.

## 5. Conclusions 

In the current study, we successfully targeted dysregulated CTNNA1, CTNNB1, TLN1, VCL, PXN, and ACTN1 genes by delivering respective siRNAs with the help of nano-sized CA particles in breast cancer cells as well as in a murine breast cancer model. In the murine metastatic cell line (4T1), significant inhibition was observed in AKT and MAPK pathways, which indicates that, by targeting these selective genes, the essential pathways involved in tumor cell proliferation could be affected. We showed downregulation of both phosphorylated and total AKT in samples treated with CA-associated siRNAs targeting CTNNA1, CTNNB1, TLN1, VCL, PXN, and ACTN1 mRNA transcripts and downregulation of phosphorylated MAPK and total MAPK in all of the samples treated with CTNNA1, CTNNB1, and VCL mRNAs. Since AKT and MAPK are downstream signaling molecules of the targeted adhesion molecules playing a critical role in oncogenesis, altercation in their expression and activation levels compared to controls indicate successful knockdown of CTNNA1, CTNNB1, TLN1, VCL, PXN, and ACTN1 target adhesion molecules. The delivery of CA-siRNA complexes caused significant tumor inhibitory effects even at a low concentration of siRNAs, likely as a result of low pH responsive-dissolution of the particles and fast endosomal release of the siRNAs from the nanoparticle carrier complex. Figure 18 summarizes the association of various cell adhesion molecules and their role in adhesion, proliferation, and metastasis of tumor cells and how CA-siRNA targeted delivery can result in cleavage of mRNA sequences of overregulated genes, which results in the inhibition of essential proliferation pathways. Thus, targeting of these dysregulated genes via RNAi and by use of the CA delivery vehicle could confer promising therapeutic benefit to human malignancy such as breast cancer.

## Figures and Tables

**Figure 1 pharmaceutics-11-00309-f001:**
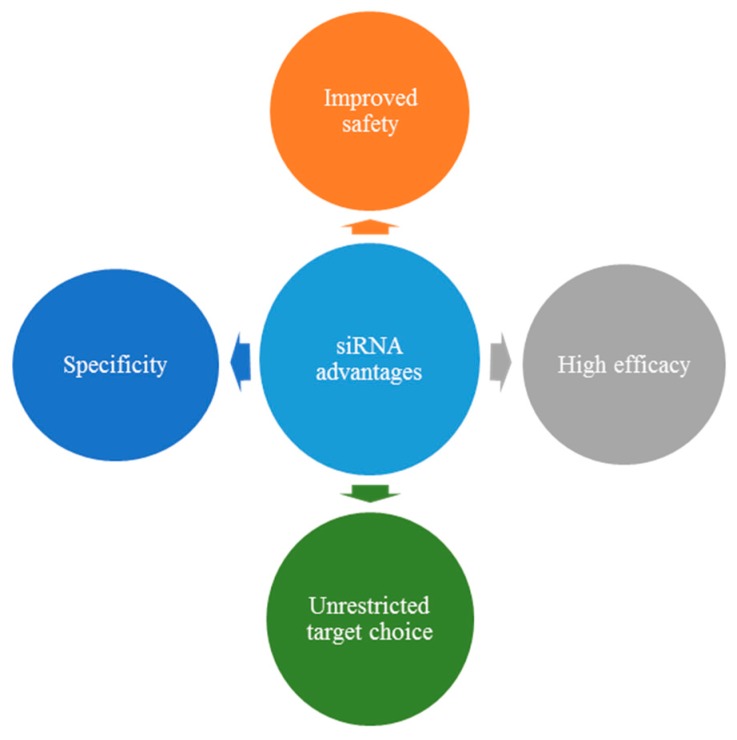
Advantages of using siRNAs as therapeutics for treatment of diseases such as breast cancer.

**Figure 2 pharmaceutics-11-00309-f002:**
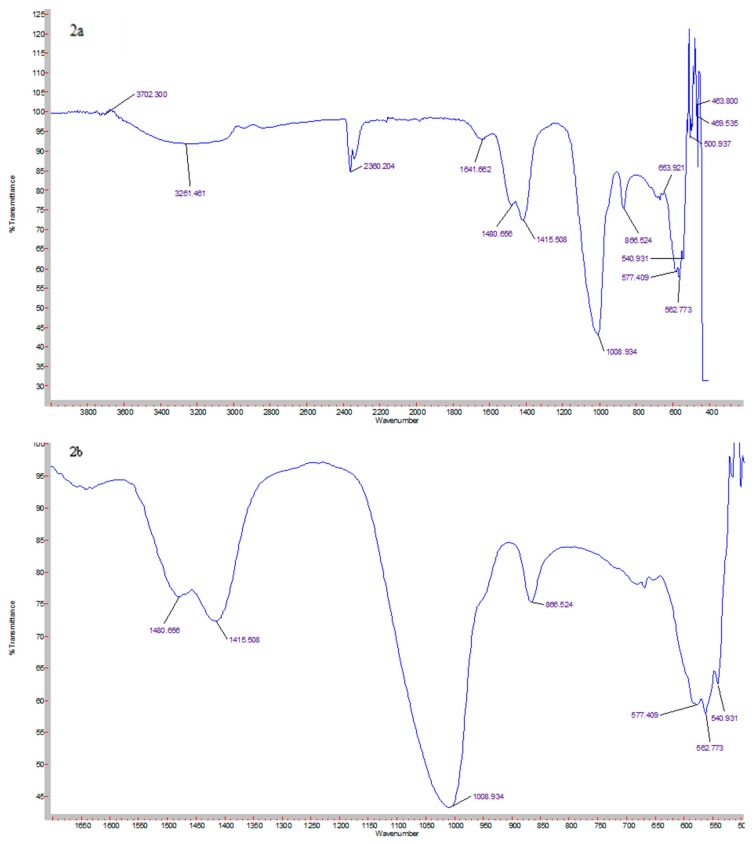
FT-IR spectra of lyophilized carbonate apatite (CA): (**a**) Spectra in the range of 400–3800 cm^−1^, and (**b**) magnified peaks of CO_3_^−^ and PO_4_^3−^.

**Figure 3 pharmaceutics-11-00309-f003:**
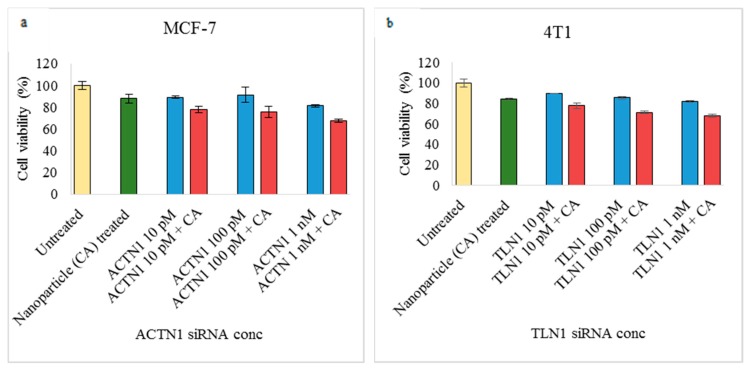
Cell viability of MCF-7 cells and 4T1 cells via the MTT assay. Cells were treated with/without CA bound with (**a**) actinin-1 (ACTN1) and (**b**) talin-1 (TLN1) siRNA at 10 pM, 100 pM, and 1 nM concentration of siRNAs for 48 h. Transfection of this complex was done for 48 h, which was followed by absorbance reading at 595 nm with a reference wavelength of 650 nm. Data is presented as mean ± S.D.

**Figure 4 pharmaceutics-11-00309-f004:**
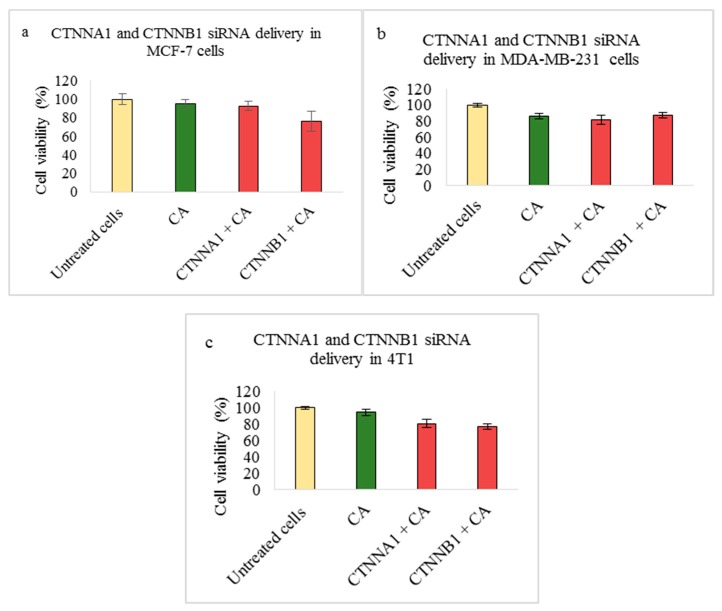
Effect of NPs-loaded siRNA against catenin alpha 1 (CTNNA1) and catenin beta 1 (CTNNB1) gene on cell viability after 48 h using the MTT assay. CA-CTNNA1 and CA-CTNNB1 complexes were prepared at a final volume of 1 mL by combining 1 nM CTNNA1 and CTNNB1 siRNAs individually along with 4 mM CaCl_2_ into DMEM buffered with bicarbonate, at a pH of 7.4, followed by a 30-min incubation. Transfection of this complex was done for 48 h, which was followed by an absorbance reading at 595 nm with a reference wavelength of 650 nm. Data is presented as mean ± S.D against (**a**) MCF-7 cells, (**b**) MDA-MB-231 cells, and (**c**) 4T1 cells.

**Figure 5 pharmaceutics-11-00309-f005:**
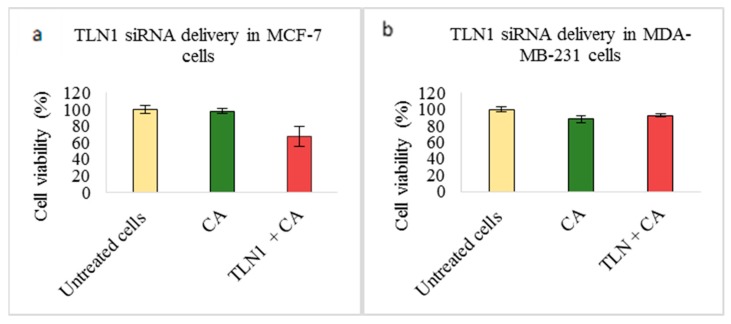
Effect of NPs-loaded siRNA against talin-1 (TLN1) gene on cell viability after 48 h using the MTT assay. CA-TLN1 complex were prepared at a final volume of 1 mL by combining 1 nM TLN1 siRNA along with 4 mM CaCl_2_ into DMEM buffered with bicarbonate, at a pH of 7.4, which was followed by a 30-min incubation. Transfection of this complex was done for 48 h, which was followed by an absorbance reading at 595 nm with a reference wavelength of 650 nm. Data is presented as mean ± S.D against (**a**) MCF-7 cells, (**b**) MDA-MB-231 cells, and (**c**) 4T1 cells.

**Figure 6 pharmaceutics-11-00309-f006:**
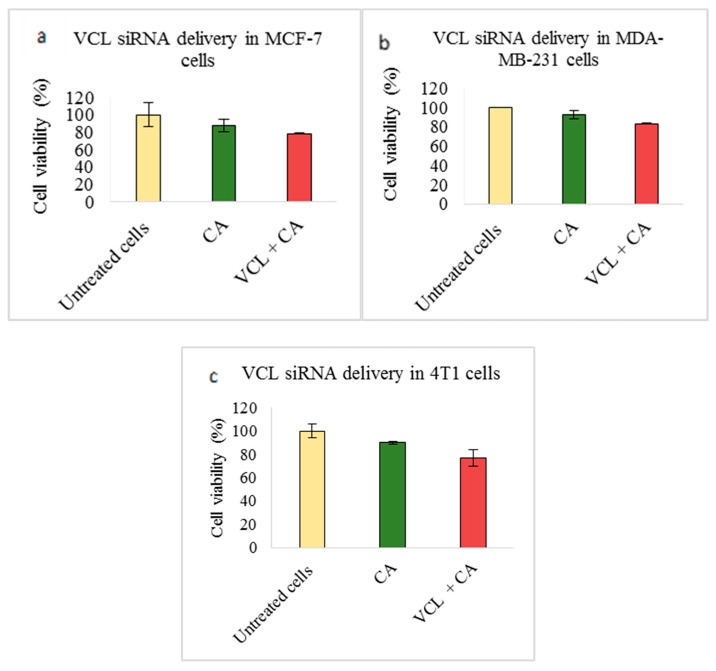
Effect of NPs-loaded siRNA against the vinculin (VCL) gene on cell viability after 48 h using an MTT assay. The CA-VCL complex was prepared at a final volume of 1 mL by combining 1 nM VCL siRNA along with 4 mM CaCl_2_ into DMEM buffered with bicarbonate, at a pH of 7.4, followed by a 30-min incubation. Transfection of this complex was done for 48 h. This was followed by an absorbance reading at 595 nm with a reference wavelength of 650 nm. Data is presented as mean ± S.D against (**a**) MCF-7 cells, (**b**) MDA-MB-231 cells, and (**c**) 4T1 cells.

**Figure 7 pharmaceutics-11-00309-f007:**
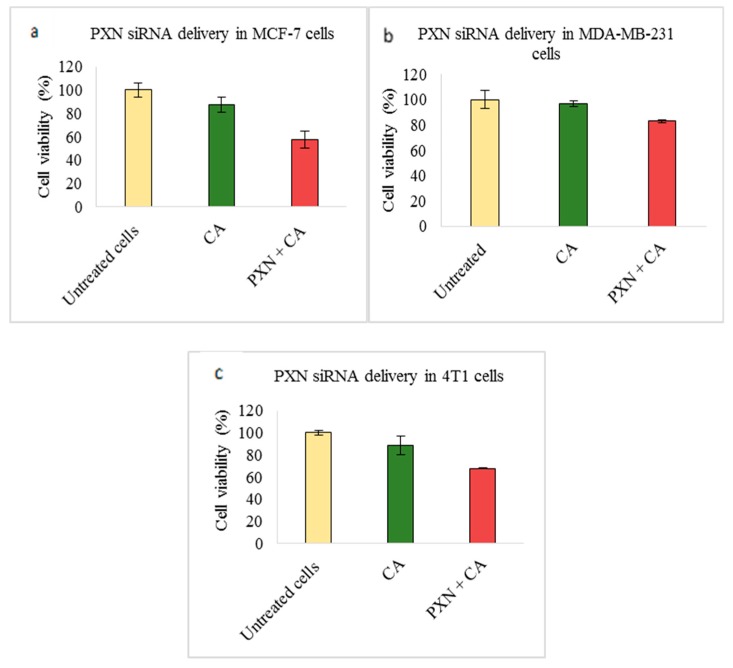
Effect of NPs-loaded siRNA against the paxillin (PXN) gene on cell viability after 48 h using the MTT assay. The CA-PXN complex was prepared at a final volume of 1 mL by combining 1 nM PXN siRNA along with 4 mM CaCl_2_ into DMEM buffered with bicarbonate, at a pH of 7.4, which was followed by a 30-min incubation. Transfection of this complex was done for 48 h, which was followed by an absorbance reading at 595 nm with a reference wavelength of 650 nm. Data is presented as mean ± S.D against (**a**) MCF-7 cells, (**b**) MDA-MB-231 cells, and (**c**) 4T1 cells.

**Figure 8 pharmaceutics-11-00309-f008:**
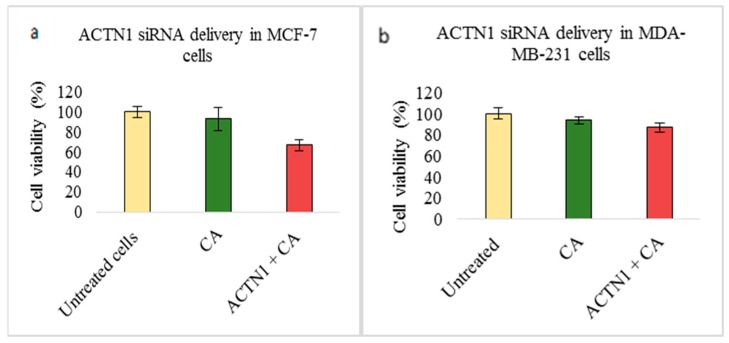
Effect of NPs-loaded siRNA against actinin-1 (ACTN1) gene on cell viability after 48 h using the MTT assay. The CA-ACTN1 complex was prepared at a final volume of 1 mL by combining 1 nM ACTN1 siRNA along with 4 mM CaCl_2_ into DMEM buffered with bicarbonate, at a pH of 7.4, which was followed by 30 min of incubation. Transfection of this complex was done for 48 h, which was followed by an absorbance reading at 595 nm with a reference wavelength of 650 nm. Data is presented as mean ± S.D against (**a**) MCF-7 cells, (**b**) MDA-MB-231 cells, and (**c**) 4T1 cells.

**Figure 9 pharmaceutics-11-00309-f009:**
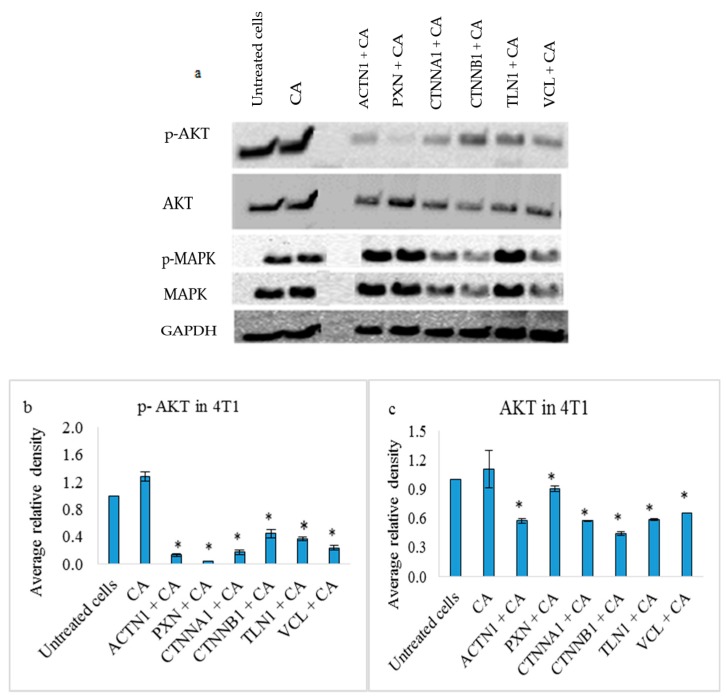
(**a**) Effect of intracellular delivery of CA loaded single additional cell adhesion siRNAs on protein expressions in 4T1 cells. Cells were incubated with CA loaded single additional cell adhesion siRNAs (ACTN1, PXN, CTNNA1, CTNNB1, TLN1, and VCL) siRNAs for 48 h, which was followed by cell lysis for Western blot analysis. After loading uniform (9 µg) concentration of proteins on SDS-PAGE, proteins were transferred onto a nitrocellulose membrane for detecting the expression of p-AKT, total AKT, p-MAPK, total MAPK, and the housekeeping gene GAPDH. Densitometry analysis of (**b**) p-AKT, (**c**) AKT, (**d**) p-MAPK, and (**e**) MAPK expression in 4T1 cells treated with single cell adhesion siRNAs (ACTN1, PXN, CTNNA1, CTNNB1, TLN1, and VCL) loaded CA. Data represents as mean ± S.D. * represents a significant difference of single siRNAs compared to CA, with *p* < 0.05.

**Figure 10 pharmaceutics-11-00309-f010:**
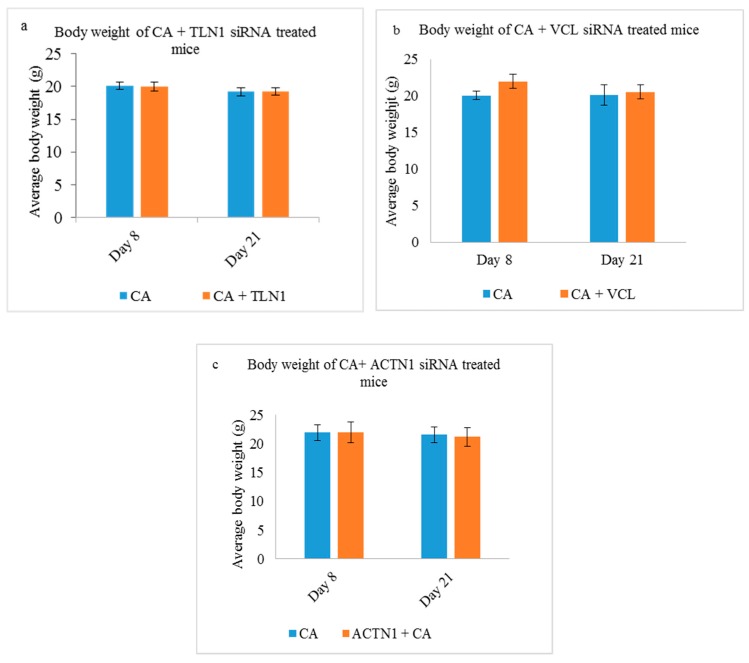
Average body weight of CA treated (**a**) CA + TLN1 (**b**) CA + VCL and (**c**) CA+ ACTN1 group of mice on Day 8 and Day 21 of the tumor regression study. Data represents an average body weight of five mice ± S.D.

**Figure 11 pharmaceutics-11-00309-f011:**
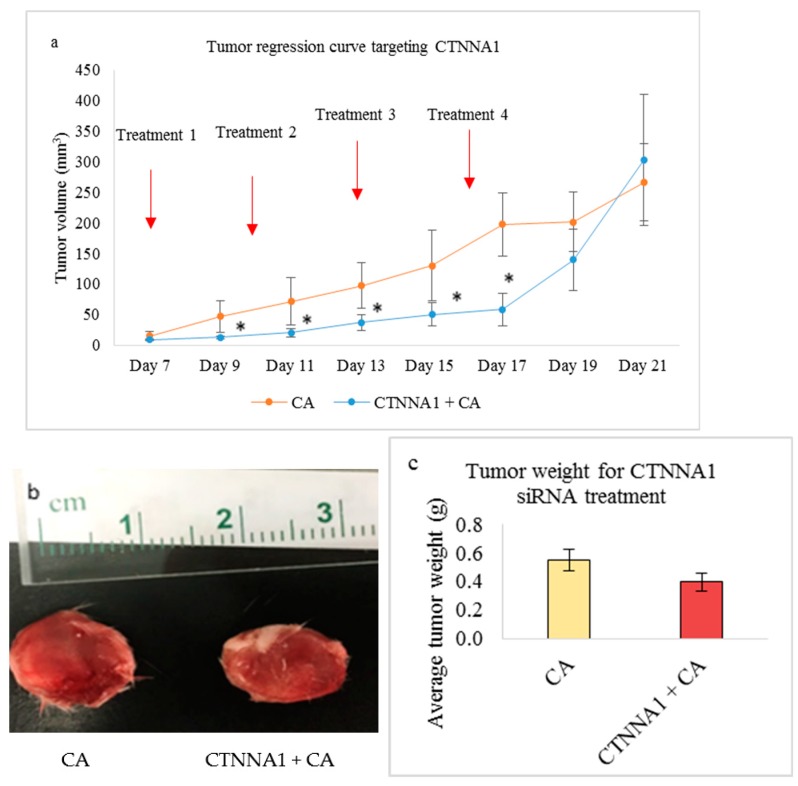
Treatment effect of CA bound siRNA against CTNNA1 genes in the 4T1 induced tumor mice model. 4T1 cells were injected subcutaneously in the mammary pad of mice. After a palpable tumor, mice were treated intravenously through tail vein injection with 100 μL of CA-CTNNA1 siRNA formed in 4 μL of 1 M CaCl_2_. Four doses of treatment were given on Day 8, Day 11, Day 14, and Day 17. Five mice were used per group and data was represented as mean ± SD. (**a**) Tumor outgrowth of mice treated with CA-CTNNA1 values are significant for * representing *p* < 0.05 compared to the CA control. (**b**) Excised tumor of the CA-CTNNA1 siRNA group compared to CA at the end of the treatment. (**c**) Tumor weight of the treated group vs. the CA-treated group.

**Figure 12 pharmaceutics-11-00309-f012:**
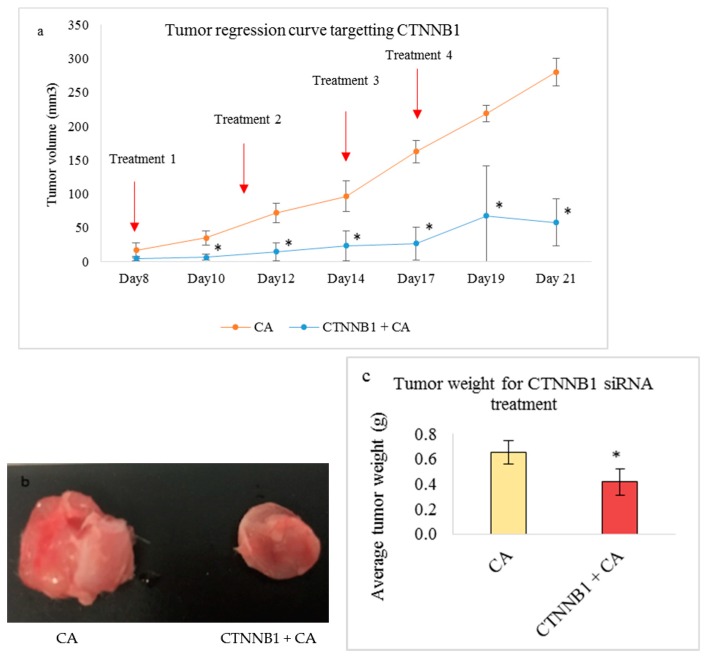
Treatment effect of CA bound siRNA against CTNNB1 genes in a 4T1 induced tumor mice model. 4T1 cells were injected subcutaneously in the mammary pad of mice. After a palpable tumor, mice were treated intravenously through tail vein injection with 100 μL of CA-CTNNB1 siRNA formed in 4 μL of 1 M CaCl_2_. Four doses of treatment were given on Day 8, Day 11, Day 14, and Day 17. Five mice were used per group and data was represented as mean ± SD. (**a**) Tumor outgrowth of mice treated with CA-CTNNB1 values are significant for * representing *p* < 0.05 compared to the CA control. (**b**) Excised tumor of the CA-CTNNB1 siRNA group compared to CA at the end of the treatment. (**c**) Tumor weight of the treated group vs. the CA treated group with * representing *p* < 0.05 compared to the control.

**Figure 13 pharmaceutics-11-00309-f013:**
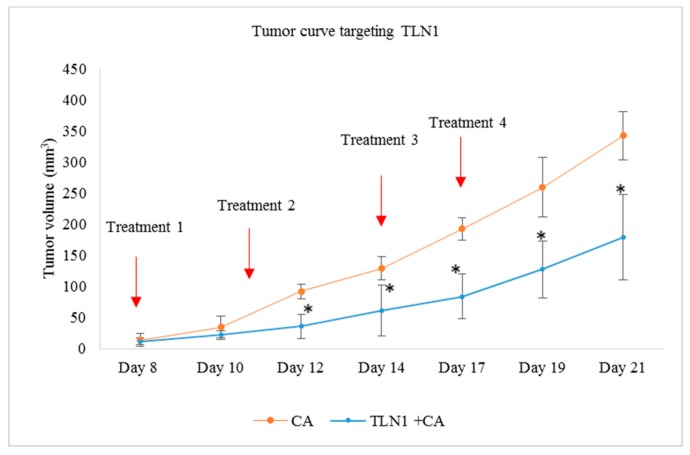
Treatment effect of CA bound siRNA against TLN1 genes in a 4T1 induced tumor mice model. 4T1 cells were injected subcutaneously in the mammary pad of mice. After a palpable tumor, mice were treated intravenously through tail vein injection with 100 μL of CA-TLN1 siRNA formed in 4 μL of 1 M CaCl_2_. Four doses of treatment were given on Day 8, Day 11, Day 14, and Day 17. Five mice were used per group and data was represented as mean ± SD. (**a**) Tumor outgrowth of mice treated with CA-TLN1 values are significant for * representing *p* < 0.05 compared to the CA control. (**b**) Excised tumor of the CA-TLN1 siRNA group compared to CA at the end of the treatment. (**c**) Tumor weight of the treated group vs. the CA treated group with * representing *p* < 0.05 compared to the control.

**Figure 14 pharmaceutics-11-00309-f014:**
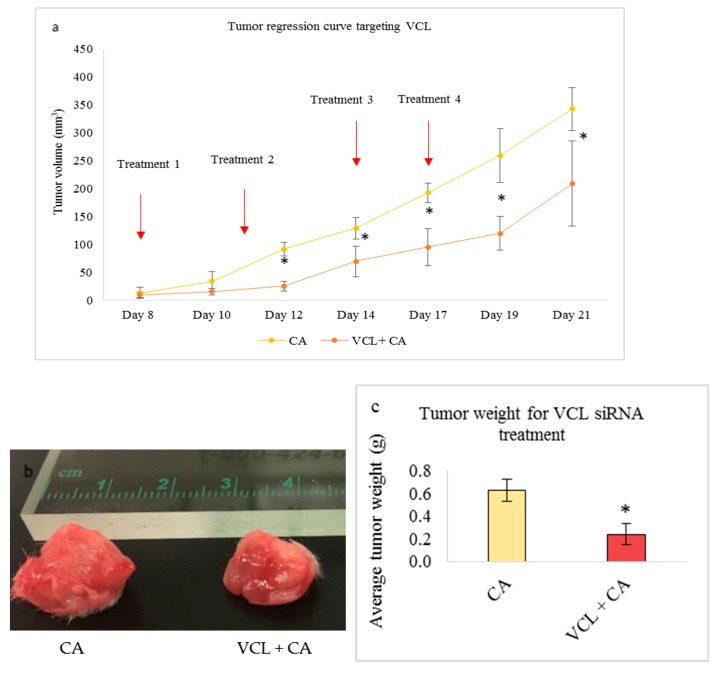
Treatment effect of CA bound siRNA against VCL genes in a 4T1 induced tumor mice model. 4T1 cells were injected subcutaneously in the mammary pad of mice. After a palpable tumor, mice were treated intravenously through tail vein injection with 100 μL of CA-VCL siRNA formed in 4 μL of 1 M CaCl_2_. Four doses of treatment were given on Day 8, Day 11, Day 14, and Day 17. Five mice were used per group and data was represented as mean ± SD. (**a**) Tumor outgrowth of mice treated with CA-VCL values are significant for * representing *p* < 0.05 compared to the CA control. (**b**) Excised tumor of the CA-VCL siRNA group compared to CA at the end of the treatment. (**c**) Tumor weight of the treated group vs. the CA-treated group with * representing *p* < 0.05 compared to the control.

**Figure 15 pharmaceutics-11-00309-f015:**
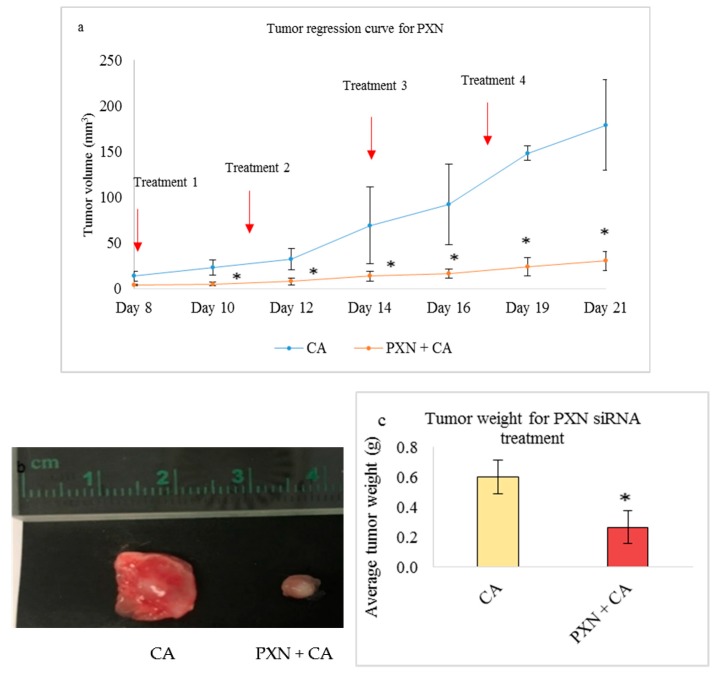
Treatment effect of CA bound siRNA against PXN genes in a 4T1 induced tumor mice model. 4T1 cells were injected subcutaneously in the mammary pad of mice. After a palpable tumor, mice were treated intravenously through tail vein injection with 100 μL of CA-PXN siRNA formed in 4 μL of 1 M CaCl_2_. Four doses of treatment were given on Day 8, Day 11, Day 14, and Day 17. Five mice were used per group and data was represented as mean ± SD. (**a**) Tumor outgrowth of mice treated with CA-PXN values are significant for * representing *p* < 0.05 compared to the CA control. (**b**) Excised tumor of the CA-PXN siRNA group was compared to CA at the end of the treatment. (**c**) Tumor weight of the treated group vs. the CA treated group with * representing *p* < 0.05 compared to the control.

**Figure 16 pharmaceutics-11-00309-f016:**
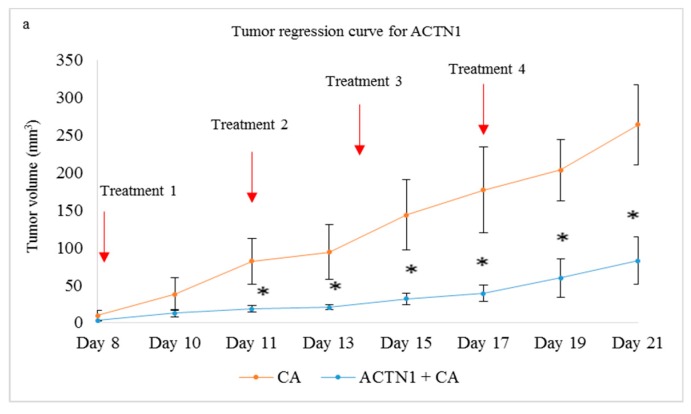
Treatment effect of CA bound siRNA against ACTN1 genes in a 4T1 induced tumor mice model. 4T1 cells were injected subcutaneously in the mammary pad of mice. After a palpable tumor, mice were treated intravenously through tail vein injection with 100 μL of CA-ACTN1 siRNA formed in 4 μL of 1 M CaCl_2_. Four doses of treatment were given on Day 8, Day 11, Day 14, and Day 17. Five mice were used per group and data was represented as mean ± SD. (**a**) Tumor outgrowth of mice treated with CA-ACTN1 values are significant for * representing *p* < 0.05 compared to the CA control. (**b**) Excised tumor of the CA-ACTN1 siRNA group was compared to CA at the end of the treatment. (**c**) Tumor weight of the treated group vs. the CA treated group with * representing *p* < 0.05 was compared to the control.

**Figure 17 pharmaceutics-11-00309-f017:**
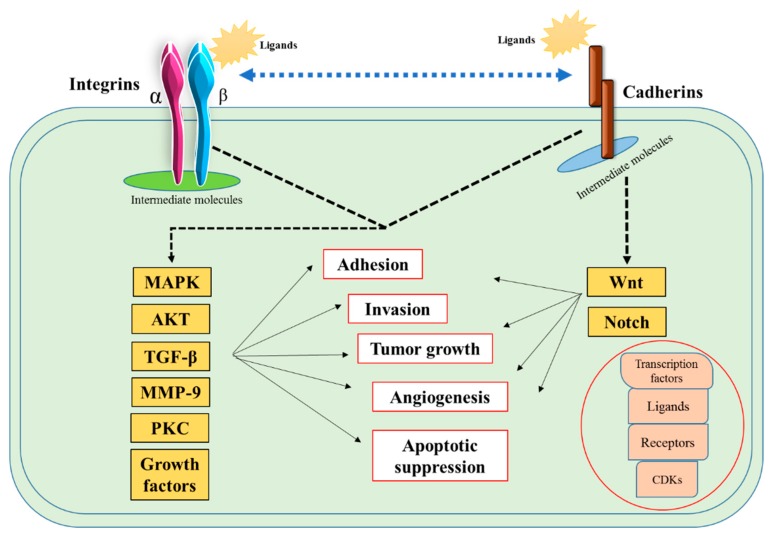
Intracellular essential signaling pathways involving cell adhesion molecules in tumorigenesis.

**Figure 18 pharmaceutics-11-00309-f018:**
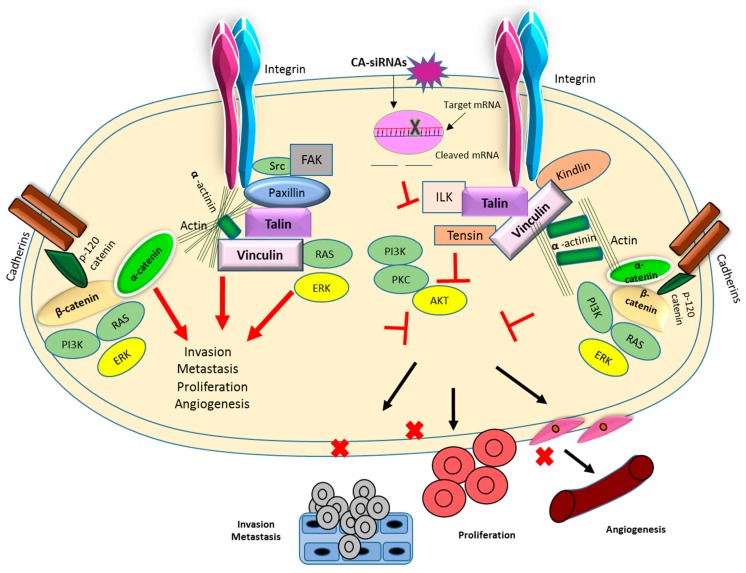
Essential cell adhesion molecules and pathways interlinked in breast tumorigenesis and aberration of these pathways and by introduction of specific siRNAs, eventually hampering tumorigenesis.

**Table 1 pharmaceutics-11-00309-t001:** siRNAs used and their sequence.

Product Name	Gene Name	Code	Target Sequence
Hs_CTNNA1_6	Catenin alpha 1	CTNNA1	AAGTGGATAAGCTGAACATTA
Hs_CTNNB1_5	Catenin beta 1	CTNNB1	CTCGGGATGTTCACAACCGAA
Hs_PXN_6	Paxillin	PXN	CCGACTGAAACTGGAACCCTT
Hs_ACTN1_5	Actinin 1	ACTN1	AACAAATCTGAATACGGCTTT
Hs_TLN1_5	Talin 1	TLN1	AACAGAGACCCCTGAAGATCC
Hs_VCL_5	Vinculin	VCL	AAAGATGATTGACGAGAGACA

**Table 2 pharmaceutics-11-00309-t002:** Enhancement of cytotoxicity of single additional cell adhesion of CA-siRNA complexes transfected in MCF-7, MDA-MB-231, and 4T1 cells.

CA + siRNAs	MCF-7		MDA-MB-231		4T1	
	Cell Viability (%) ± S.D.	Actual Cytotoxicity (%) ± S.D.	Cell Viability (%) ± S.D.	Actual Cytotoxicity (%) ± S.D.	Cell Viability (%) ± S.D.	Actual Cytotoxicity (%) ± S.D.
CTNNA1	92.65 ± 5.0	2.03 ± 6.2	81.65 ± 6.1	−2.28 ± 4.9	80.76 ± 5.4	13.42 ± 8.9
CTNNB1	76.46 ± 10.8	14.54 ± 9.3	86.93 ± 3.1	8.15 ± 7.6	77.06 ± 3.0	16.03 ± 5.3
TLN1	67.65 ± 11.8	30.2 ± 12.1	92.5 ± 1.5	−4.39 ± 2.9	66.46 ± 11.6	23.53 ± 10.4
VCL	78.34 ± 0.8	8.74 ± 7.40	83.3 ± 0.6	9.37 ± 3.9	70.67 ± 16.4	19.38 ± 16.1
PXN	57.60 ± 7.3	29.76 ± 0.8	82.92 ± 1.2	13.82 ± 3.1	67.65 ± 0.5	27.07 ± 8.9
ACTN1	66.67 ± 5.2	25.97 ± 8.9	87.15 ± 4.2	6.59 ± 0.4	71.81 ± 3.4	18.54 ± 6.3

**Table 3 pharmaceutics-11-00309-t003:** Tumor reduction in the CA-loaded CTNNA1 siRNA treated group (in folds) compared to the CA-treated group.

Day 9	Day 11	Day 13	Day 15	Day 17	Day 19	Day 21
3.74	3.41	2.65	2.60	3.40	1.44	0.88

**Table 4 pharmaceutics-11-00309-t004:** Tumor reduction in CA loaded CTNNB1 siRNA treated group (in folds) compared to the CA-treated group.

Day 10	Day 11	Day 13	Day 15	Day 17	Day 19	Day 21
4.8	3.8	3.7	5.1	2.7	4.2	4.8

**Table 5 pharmaceutics-11-00309-t005:** Tumor reduction in the CA-loaded TLN1 siRNA treated group (in folds) compared to the CA-treated group.

Day 10	Day 12	Day 14	Day 16	Day 19	Day 21
1.58	2.58	2.11	2.29	2.04	1.91

**Table 6 pharmaceutics-11-00309-t006:** Tumor reduction in the CA-loaded VCL siRNA treated group (in folds) compared to the CA-treated group.

Day 10	Day 12	Day 14	Day 16	Day 19	Day 21
2.18	3.61	1.85	2.02	2.16	1.64

**Table 7 pharmaceutics-11-00309-t007:** Tumor reduction in the CA-loaded PXN siRNA treated group (in folds) compared to the CA treated group.

Day 10	Day 12	Day 14	Day 16	Day 19	Day 21
4.80	4.02	5.02	5.61	6.19	5.92

**Table 8 pharmaceutics-11-00309-t008:** Tumor reduction in the CA-loaded ACTN1 siRNA treated group (in folds) compared to the CA treated group.

Day 10	Day 11	Day 13	Day 15	Day 17	Day 19	Day 21
3.02	4.42	4.64	4.49	4.50	3.42	3.19

**Table 9 pharmaceutics-11-00309-t009:** Critical signaling pathways involved in various breast cancers.

Proteins	Critical Signaling Pathways	Expression in	References
Catenins	WNT	Adenocarcinomas, TNBC/Basal-like breast cancers, Metastasis	[48,49]
Talin	P13K-AKT, RAS, FAK-MAPK	TNBC/Basal-like breast cancers, HER2-overexpressing, Metastasis	[13,50,51,52]
Vinculin	FAK-MAPK, PI3K	TNBC	[14,21,53,54]
Paxillin	Src/FAK/PI3K, ERK, Rho/ROCK, JNK, p38 MAPK	Adenocarcinomas, TNBC/Basal-like breast cancers, HER2 overexpressing, Metastasis	[7,8,22,55,56]
Actinin	P13K-AKT, MAPK	Adenocarcinomas, TNBC/Basal-like breast cancers	[9,10,18,57]

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
