# Peer review of "Targeting Cell Adhesion Molecules via Carbonate Apatite-Mediated Delivery of Specific siRNAs to Breast Cancer Cells In Vitro and In Vivo"

_pharmaceutics, 2019, doi:10.3390/pharmaceutics11070309_

Reviewer 1 Report

Ezharul Hoque Chowdhury and co-workers aimed at using carbonate apatite to deliver specific siRNA in in vitro and in vivo breast cancer models. 

Major revision

1) Introduction must be improved. Authors should highlight the problem that they want to solve with their research.

2) The use of nanoparticles in breast cancer and drug delivery (e.g extracellular vesicles and liposomes) should be reported. A proper review of the literature must be performed.

3) Size and morphology of produced nanoparticles should be provided (cry-EM, AFM)

4) Uptake study should be performed both in vitro and in vivo

5) Transcriptomic analysis (RNA-seq) would be needed to explain the mechanism

Author Response

Reviewer 1

1)    Introduction must be improved. Authors should highlight the problem that they want to solve with their research.

We have improved the introduction and have also highlighted the research gap which we are focusing on towards the end of the introduction. In addition, we have also added some diagrams in introduction as well as in discussion section (Figure 1, Figure 17 and Figure 18).

2)    The use of nanoparticles in breast cancer and drug delivery (e.g extracellular vesicles and liposomes) should be reported. A proper review of the literature must be performed.

We have added information about various nanoparticles and how they are being used as delivery vehicles in breast cancer.

3)    Size and morphology of produced nanoparticles should be provided (cry-EM, AFM)

We have published the results on particle size distribution, size, morphology based on DLS, TEM, SEM for nanoparticles in our following previous papers:

Hossain S, Stanislaus A, Chua MJ, Tada S, Tagawa Y, et al. (2010) Carbonate apatite facilitated intracellularly delivered siRNA for efficient knockdown of functional genes. J Control Release 147: 101-108

“Tiash S, Kamaruzman NI, Chowdhury EH, (2017), Carbonate Apatite Nanoparticles Carry siRNA(s) Targeting Growth Factor Receptor Genes, EGFR1 and ERBB2 to Regress Mouse Breast Tumor, Drug Delivery, 24(1), 1721-1730

Kamaruzman, N., Tiash, S., Ashaie, M., & Chowdhury, E. (2018). siRNAs Targeting Growth Factor Receptor and Anti-Apoptotic Genes Synergistically Kill Breast Cancer Cells through Inhibition of MAPK and PI-3 Kinase Pathways. Biomedicines, 6(3), 73.

We have included another section in this paper, section 2.2.3 where we have put our FT-IR methodology and section 3.1 where we have shown the FT-IR results as a part of characterization of our nanoparticles. We have discussed about it in discussion as well.

4)    Uptake study should be performed both in vitro and in vivo

We have previously shown uptake results in our following paper:

Tiash, S., & Chowdhury, E. H. (2019). siRNAs targeting multidrug transporter genes sensitise breast tumour to doxorubicin in a syngeneic mouse model. Journal of drug targeting, 1-13.

Tiash S, Kamaruzman NI, Chowdhury EH, (2017), Carbonate Apatite Nanoparticles Carry siRNA(s) Targeting Growth Factor Receptor Genes, EGFR1 and ERBB2 to Regress Mouse Breast Tumor, Drug Delivery, 24(1), 1721-1730

Tiash, S., & Chowdhury, E. H. (2016). Passive targeting of cyclophosphamide-loaded carbonate apatite nanoparticles to liver impedes breast tumor growth in a syngeneic model. Current pharmaceutical design22(37), 5752-5759.

 5) Transcriptomic analysis (RNA-seq) would be needed to explain the mechanism

We have used scrambled siRNA as a control in our previously published papers, showing no antitumor effects for both in vitro and in vivo.(1) We have also used siRNAs which were functionally validated by qRT-PCT with more than 80% knockdown efficacy. Further, we have done Western blot to see the effect of these CA-siRNA combinations, and based on our results we have shown that these genes could be correlated to downstream molecules such as MAPK and AKT (total and phosphorylated form) which play essential roles in tumor proliferation and metastasis pathways and by regulating the gene expression of these essential cell adhesion molecules, we can regulate the expression of MAPK and AKT.

1.Tiash S, Kamaruzman NIB, Chowdhury EH. Carbonate apatite nanoparticles carry siRNA (s) targeting growth factor receptor genes egfr1 and erbb2 to regress mouse breast tumor. Drug delivery. 2017;24(1):1721-30.

Reviewer 2 Report

Authors, in the submitted paper, use the RNA interference technology to silence a number of genes involved in pathogenesis of breast cancer.

There are some major and minor concerns:

Major concerns

line 108: 2.2.2: siRNA design and sequence and line 540: Table 1:

There is no information how the targeted sequences selected ? via some algorithm, via some other relative reference?

Does specificity of siRNAs has been checked?

How did the authors confirm the successful knock down of the corresponding genes for each siRNA ? with RT-PCR ? with q-PCR ? western blot analysis to see the corresponding protein levels?

To what extent the selected siRNAs knock downed the targeted genes?

Is there any preliminary data that are not mentioned here?

lines 209-213, 328-329, 392-393 and 601-608 (Fig.7)

Authors must normalize the intensity bands via the use of a software, like the Image J or image studio lite. There is a discrepancy / misinterpretation between the Figure 7, the results and the resulting discussion, and normalization will help clarify this issue.

Minor concerns

line 125: 

in what volume seeded cells were treated with siRNA-CA complexes

lane 561:  Fig.2

does bar, corresponding to CTNNB-CA, should have a different color, to be in accordance with Fig.2.b and Fig.2.c  ?

Authors, in conclusion section, should also correlate their results, to point out the best expected results, via RNA interference of which genes, and then, could comment whether co-dysregulation of which genes based on their results and the involved mechanism could give a much better result concerning the reduction of the tumor growth.

Author Response

Reviewer 2

There are some major and minor concerns:

Major concerns

line 108: 2.2.2: siRNA design and sequence and line 540: Table 1:

There is no information how the targeted sequences selected ? via some algorithm, via some other relative reference?

We have used the siRNAs that were functionally validated by Qiagen by qRT-PCR with more than 80% knockdown efficacy.

Does specificity of siRNAs has been checked?

How did the authors confirm the successful knock down of the corresponding genes for each siRNA ? with RT-PCR ? with q-PCR ? western blot analysis to see the corresponding protein levels?

To what extent the selected siRNAs knock downed the targeted genes?

Is there any preliminary data that are not mentioned here?

lines 209-213, 328-329, 392-393 and 601-608 (Fig.7)

We have used the siRNAs that were functionally validated by Qiagen by qRT-PCR with more than 80% knockdown efficacy. siRNAs have been designed based on the conserved sequences of the respective target genes, present in both human and mouse.

Authors must normalize the intensity bands via the use of a software, like the Image J or image studio lite. There is a discrepancy / misinterpretation between the Figure 7, the results and the resulting discussion, and normalization will help clarify this issue.

Thank you for highlighting the misinterpretation of the Western blot results. We had mislabelled the bands. We have corrected the labelling and also have included the results from densitometry analysis which we did using ImageJ software (Figure 9b-9e). We have normalized the treatment groups with respect to CA treated control in order to identify the significant difference between the CA control and CA-siRNA treated groups.

Minor concerns

line 125: 

in what volume seeded cells were treated with siRNA-CA complexes

We have mentioned the cell seeding density at 50,000 cells/well (Line 171).

lane 561:  Fig.2

does bar, corresponding to CTNNB-CA, should have a different color, to be in accordance with Fig.2.b and Fig.2.c  ?

Thank you for highlighting the graphs. We have rectified the mistake and changed the colour of the CTNNB-CA graph in accordance with other two figures.

Round  2

Reviewer 1 Report

Authors significantly modify the manuscript which is now suitable for publication

Author Response

Thank you very much.

Reviewer 2 Report

Paper submitted is really improved.

CONCERNING A MAJOR CONCERN, however,

“Major concerns line 108: 2.2.2: siRNA design and sequence and line 540: Table 1: Does specificity of siRNAs has been checked? How did the authors confirm the successful knock down of the corresponding genes for each siRNA ? with RT-PCR ? with q-PCR ? western blot analysis to see the corresponding protein levels? To what extent the selected siRNAs knock downed the targeted genes? Is there any preliminary data that are not mentioned here? lines 209-213, 328-329, 392-393 and 601-608 (Fig.7)

Answer by authors. We have used the siRNAs that were functionally validated by Qiagen by qRT-PCR with more than 80% knockdown efficacy. siRNAs have been designed based on the conserved sequences of the respective target genes, present in both human and mouse.”

THE AUTHORS, in their revised paper,

in line 143, they say: All the siRNAs were validated by Qiagen. Table 1 lists the siRNAs used and their sequences.

THIS ANSWER IS NOT ENOUGH.

AUTHORS HAVE TO PROVE IN THEIR OWN CELL LINES THAT the used siRNAs ARE indeed FUNCTIONAL, using RT-PCR or q-PCR or western blot analysis.

Author Response

We have done Western blot (Fig. 9a and 9b) which clearly shows downregulation of both phosphorylated AKT and total AKT in all of the samples treated with nanoparticles-associated siRNAs targeting CTNNA1, CTNNB1, TLN1, VCL, PXN and ACTN1 mRNA transcripts and additionally, downregulation of phosphorylated MAPK and total MAPK in all of the samples treated with CTNNA1, CTNNB1  and  VCL  mRNAs. Since AKT and MAPK are downstream signaling molecules of the targeted adhesion molecules which play essential role in oncogenesis, changes in their expression and activation levels compared to controls are clear indicators of knockdown of CTNNA1, CTNNB1, TLN1, VCL, PXN and ACTN1. Moreover, since AKT and MAPK pathways are predominantly responsible for regulating proliferation and survival of cancer cells, downregulation of the pathways also correlate with the induced cytotoxicity of the breast cancer cells.

We have also presented the data with a scramble (negative control siRNA) in earlier studies [1,2] demonstrating no changes in cytotoxicity and tumor regression following the treatment, indicating that the enhancement in cytotoxicity and  tumor regression was not due to the non-specific interactions of the siRNA with other mRNA transcripts or any other off-target effects of the particular siRNA. In addition, we showed the evidence in our previous study [3] that carbonate apatite is highly efficient in knockdown of target genes even with pico gram amount of initially siRNA concentration.

Kamaruzman, N., Tiash, S., Ashaie, M., & Chowdhury, E. (2018). siRNAs Targeting Growth Factor Receptor and Anti-Apoptotic Genes Synergistically Kill Breast Cancer Cells through Inhibition of MAPK and PI-3 Kinase Pathways. Biomedicines, 6(3), 73.

Tiash S, Kamaruzman NI, Chowdhury EH, (2017), Carbonate Apatite Nanoparticles Carry siRNA(s) Targeting Growth Factor Receptor Genes, EGFR1 and ERBB2 to Regress Mouse Breast Tumor, Drug Delivery, 24(1), 1721-1730

Hossain, S., Stanislaus, A., Chua, M. J., Tada, S., Tagawa, Y. I., Chowdhury, E. H., & Akaike, T. (2010). Carbonate apatite-facilitated intracellularly delivered siRNA for efficient knockdown of functional genes. Journal of Controlled Release147(1), 101-108.

Round  3

Reviewer 2 Report

Reviewer 2

3rd review

On line 362, authors referred to their previous work (Ref 45), where in the corresponding Table 1,

some information is given for the used siRNAs, concerning validated cell line and percentage of knock down effect.

In this work,

even this information is missing, as shown below.

I cannot understand, while authors have done so much work, they bypassed such initial experiments to prove in their own cell lines that the used siRNAs are indeed functional.

Author Response

Thank you very much for all of the valuable comments and suggestions.